Marine heatwave challenges solutions
to human–wildlife conflict. *Proc. R. Soc. B*
**288**: 20211607.

biological applications, environmental science,
ecology

trade-offs, dynamic ocean management,
Dungeness crab, whale bycatch, marine
heatwave

**Author for correspondence:**
Jameal F. Samhouri
e-mail: jameal.samhouri@noaa.gov

Electronic supplementary material is available
online at https://doi.org/10.6084/m9.figshare.
c.5707277.

# Marine heatwave challenges solutions to human–wildlife conflict

Jameal F. Samhouri[1], Blake E. Feist[1], Mary C. Fisher[1,2], Owen Liu[3], Samuel M. Woodman[4], Briana Abrahms[5,6], Karin A. Forney[7,8], Elliott L. Hazen[5], Dan Lawson[9], Jessica Redfern[7,10] and Lauren E. Saez[11]

[1]Conservation Biology Division, Northwest Fisheries Science Center, National Marine Fisheries Service, National Oceanic and Atmospheric Administration, Seattle, WA, USA
[2]School of Environmental and Forest Sciences, University of Washington, Seattle, WA, USA
[3]NRC Research Associateship Program, Northwest Fisheries Science Center, National Marine Fisheries Service, National Oceanic and Atmospheric Administration, Seattle, WA, USA
[4]Ocean Associates, Inc., under contract to Marine Mammal and Turtle Division, Southwest Fisheries Science Center, National Marine Fisheries Service, National Oceanic and Atmospheric Administration, La Jolla, CA, USA
[5]Environmental Research Division, Southwest Fisheries Science Center, National Marine Fisheries Service, National Oceanic and Atmospheric Administration, Monterey, CA, USA
[6]Department of Biology, Center for Ecosystem Sentinels, University of Washington, Seattle, WA, USA
[7]Marine Mammal and Turtle Division, Southwest Fisheries Science Center, National Marine Fisheries Service, National Oceanic and Atmospheric Administration, Moss Landing, CA, USA
[8]Moss Landing Marine Laboratories, San Jose State University, Moss Landing, CA, USA
[9]Protected Resources Division, West Coast Regional Office, National Marine Fisheries Service, National Oceanic and Atmospheric Administration, Long Beach, CA, USA
[10]Anderson Cabot Center for Ocean Life, New England Aquarium, Boston, MA, USA
[11]Ocean Associates, Inc., under contract to Protected Resources Division, West Coast Regional Office, National Marine Fisheries Service, National Oceanic and Atmospheric Administration, Long Beach, CA, USA

JFS, 0000-0002-8239-3519; SMW, 0000-0001-6071-8186; LES, 0000-0003-4988-4655

Despite the increasing frequency and magnitude of extreme climate events, little is known about how their impacts flow through social and ecological systems or whether management actions can dampen deleterious effects. We examined how the record 2014–2016 Northeast Pacific marine heatwave influenced trade-offs in managing conflict between conservation goals and human activities using a case study on large whale entanglements in the U.S. west coast's most lucrative fishery (the Dungeness crab fishery). We showed that this extreme climate event diminished the power of multiple management strategies to resolve trade-offs between entanglement risk and fishery revenue, transforming near win–win to clear win–lose outcomes (for whales and fishers, respectively). While some actions were more cost-effective than others, there was no silver-bullet strategy to reduce the severity of these trade-offs. Our study highlights how extreme climate events can exacerbate human–wildlife conflict, and emphasizes the need for innovative management and policy interventions that provide ecologically and socially sustainable solutions in an era of rapid environmental change.

## 1. Introduction

Extreme climate events wreak havoc on social and ecological systems [1,2], and these threats are growing in frequency and magnitude [3]. In recent decades, tropical cyclones caused USD 2.1T in damage worldwide [4], wildfires destroyed an average of 3 million ha annually in the United States alone [5], and extreme temperatures affected nearly 100 million people globally [6]. Yet relatively little is known about how climate extremes influence links between biophysical and human components of ecosystems, even though there is strong potential for them to exacerbate conflict between people and wildlife [7]. Heatwaves—discrete and prolonged warming events [2]—are prominent

examples, as they can alter proximity of wildlife to areas used by people for food production. These ecological responses heighten societal concerns related to public safety, competition (for crops or fisheries), predation and incidental capture [7,8]. For instance, heatwave-associated drought is known to alter the behaviour and distribution of terrestrial mammals, leading them to cause massive damage to crops and livestock [9]. Even less well understood than the intensification of such conflicts is how interventions designed to mitigate effects of climate extremes may have unintended costs for society and ecosystems. However, severe environmental conditions have the potential to fundamentally alter the effectiveness of management strategies attempting to balance benefits and costs, or trade-offs, within social–ecological systems [10].

In the ocean, heatwaves are intensified by climate change and are emerging globally as a particularly vexing environmental concern [2,11]. A marine heatwave is defined as a 'prolonged discrete anomalously warm water event that can be described by its duration, intensity, rate of evolution and spatial extent' [12]. Heatwaves reorganize ecological communities by thermally displacing preferred habitat, causing mass wildlife mortalities and strandings, and provoking outbreaks of diseases and nuisance species [13–15]. Biophysical changes driven by heatwaves in turn produce social and economic changes, such as shifts in fishing grounds and fisheries yields, redirection of conservation resources and disaster-management interventions intended to protect public health [16–18]. Distributional shifts of species are at the core of many of these social responses. While they can create new opportunities that benefit people directly or indirectly [10], distributional shifts can also generate strain between ocean use practices that were historically sustainable under cooler ocean conditions. For example, in many regions around the world large whales have experienced changes in the timing and pathways of migration and occurrence on feeding grounds. Some of these shifts are due to heatwaves specifically [2,18], though not all are [19,20], and can be especially problematic when altered spatial distributions or movements lead to new or increased conflict with human activities, including collisions with ships [19,21] and escalation of incidental catch in fisheries (i.e. bycatch [22]). Due to the difficulty in quantifying these dynamics at sea [23], extreme climate events can produce trade-offs not previously addressed by management strategies geared at longer-term issues.

On the US west coast, most large whale populations are increasing because of strong legal protections following the cessation of hunting in the middle of the last century [24]. However, this conservation success story has met unforeseen challenges in recent years due to a combination of changing environmental conditions, whale distributional shifts, and an increasingly crowded ocean. A marine heatwave of unprecedented extent and severity, that lasted from 2014 to 2016 [25] with physical, biological and social impacts that persisted much longer [26], provides a case in point. During and immediately following the heatwave, reports of endangered blue whales (*Balaenoptera musculus*) snared in fishing gear emerged for the first time. In addition, more than 100 entangled humpback whales (*Megaptera novaeangliae*) including some from threatened and endangered populations, were reported in just a few years, compared to approximately 50 total in the greater than three decades previously [27]. Observational evidence from central California suggested this dramatic

increase in humpback whale entanglements was due in part to an onshore shift of their feeding grounds during the heatwave [28], but to date there has been no distributional information available year-round, at broad enough spatial scales, and fine enough resolutions, to support these findings.

Most reported entanglements emerged in California. While the fishery responsible for these entanglements is often unknown [27], the Dungeness crab (*Metacarcinus magister*) fishery—the most valuable fishery on the US west coast in recent decades and which engages more than 1000 vessels— was implicated in the majority of cases where fishing gear was identified. Though this fishery is scheduled to open after most whales have migrated to breeding grounds outside the continental US and to close before the whales migrate back [29], the fishery faced extensive delays during the heatwave due to harmful algal blooms that contaminated crabs with levels of domoic acid unsafe for human consumption [17,30–32]. The delays pushed the majority of Dungeness crab fishing activity from winter into spring, disrupting the historically offset seasonal patterns of fishing and whale migration [28]. Since the initial spike in whale entanglements on the US west coast, the State of California has instituted a Risk Assessment and Mitigation Programme (RAMP) that brings together representatives from state and federal agencies, the fishing industry, conservation groups and the public to tackle this problem. However, to date there has been no information available about the spatial dynamics of the fishery to explore changing overlap between large whales and the potential source of entanglements, or viable solutions to this problem.

This specific situation off the US west coast is emblematic of many in which extreme climate events disrupt social and ecological dynamics. In many of the more well-documented cases, climate extremes amplify human–wildlife conflict [33], though this need not always be the case [10,34]. Irrespective of the outcome, quantitative information to determine if and how interventions can counteract undesired effects, and encourage desired effects, tends to be lacking. One increasingly discussed solution, dynamic management, is to increase flexibility so that managers can better adjust their actions in response to changing environmental conditions [23]. Dynamic ocean management to reduce bycatch of migratory and highly mobile species of conservation concern offers particular promise [35,36], but it remains an open question whether these strategies can consistently produce win– win outcomes for protected species (by reducing bycatch) and fisheries (by maintaining or increasing yields), or will at times result in win–lose or even lose–lose outcomes [37]. While a previous study described the physical ingredients underlying the rise in entanglements on the US west coast [28], here for the first time we (i) evaluate economic consequences of this human–wildlife conflict, (ii) quantify the spatial and temporal dynamics behind them and (iii) explore the potential for dynamic management strategies to mitigate trade-offs between whale conservation goals and the sustainability of the California Dungeness crab fishery before, during and after this period of social–ecological squeeze. By combining dynamic whale distribution models and nearly 400 000 remotely sensed geolocations of fishing vessels, we highlight the difficulty of managing this specific human–wildlife conflict in a changing climate and, more generally, underscore the need for fresh solutions that keep pace with the moving target of sustainability [38].

## 2. Methods

We conducted (i) a retrospective evaluation of changes in whale entanglement reports, entanglement risk and revenue to the California Dungeness crab fishery from 2009 to 2019 and (ii) a hindcast scenario analysis to test whether systematic restrictions to the fishery could have mitigated elevated entanglement risk during the heatwave while also avoiding substantial fisheries losses. We compared outcomes across three distinct time periods representing before (2009–2014, pre-heatwave), during (2014–2018, heatwave) and after (2018–2019, post-heatwave) biophysical changes associated with the 2014–2016 Northeast Pacific heatwave. We considered the heatwave period to extend through mid-2018 because the heatwave caused compression of cool, productive whale habitat near shore [28], and these and other effects persisted after the MHW subsided [26].[1] All analyses were performed in R [39].

For the retrospective evaluation, we quantified changes in the number of blue and humpback whales reported as entangled in California Dungeness crab fishing gear over the period 2009–2019 using the database described by Saez *et al.* [27]. Typically, an entanglement occurs because the rope connecting a crab trap to a surface buoy becomes wrapped around a whale's fluke or pectoral fin, or gets caught in the whale's mouth. Because humpback and blue whales are listed as threatened or endangered under the U.S. Endangered Species Act and were of specific concern with respect to entanglement [40], we described entanglement reports for these two species individually and lumped all other species together.

We constructed an index of risk of entanglement in Dungeness crab fishing gear for blue and humpback whales based on overlap of whale and fishing distributions. We derived predicted whale distributions from habitat suitability and habitat-based density models [18,41] and fishing distributions and revenues from vessel monitoring system (VMS) data linked to California landings receipts registering Dungeness crab [42]. The VMS data contained approximately 370 000 geolocations of greater than 280 crab vessels, which we matched to greater than 16 000 landings receipts (fishing trips). This database is the only source of information about the spatial dynamics of the fishery not self-reported (at much coarser spatial scales) by fishery participants. Despite its enormous value, until now there has been no rigorous examination of the spatial dynamics of the crab fishery using data collected autonomously and at fine spatial and temporal scales.

Both whale models were validated extensively against several independent datasets, including localized aerial surveys, shipboard marine mammal surveys and standardized whale-watching data (electronic supplementary material). While these models do not account explicitly for population growth within our study period, they accurately describe the disruptions in timing of migration and distribution during the heatwave compared to years prior. In order to align whale model outputs with fishing data on a common spatial scale, we calculated area-weighted mean values for the blue and humpback whale predictions on a 5 × 5 km grid at monthly intervals (electronic supplementary material).

For the hindcast scenario analysis, we simulated changes in entanglement risk and fishery revenue expected from several approaches recently considered or implemented by US west coast state Dungeness crab fishery managers or their advisory working groups. The scenarios fell into four broad categories of time–area restrictions that are thought to address when and where whales overlap most with fishing: delayed season openings, spring closures, spring fishing depth restrictions, and spring fishing effort reductions (table 1). We applied these scenarios to each of the 10 fishing seasons between 2009–2019 (Electronic Supplementary Material), and refer to each season using 'crab years' from November of the previous year through the following

October; the 2016 crab year corresponds to the 2015–2016 fishing season of November 2015 to October 2016. The delayed season opening scenario imposed an opening of 15 December at either of two spatial extents (statewide or central management area only). Spring season (April–July) actions began 1 April and included early closures (100% effort reductions), 50% effort reductions, and depth restrictions (closures >30 fathoms) applied statewide, within the central management area only, or within blue and humpback whale Biologically Important Areas (BIAs) [29]. We compared the simulated effects of these scenarios on entanglement risk and fishery revenue to the patterns we observed in the retrospective evaluation of historical fishing activity of VMS-equipped vessels and entanglement risk (a status quo scenario). Note that in the 2019 season the fishery closed early (15 April) statewide, due to a legal settlement [41].

For all scenarios, we determined the normalized entanglement risk to blue and humpback whales as well as revenue to the crab fishery, and the change in normalized risk and change in revenue relative to status quo, within each crab fishing season. We weighted relative (per cent) changes in risk and revenue equally, though this assumption could be adjusted [43]. We also calculated the cost-effectiveness of each scenario, by comparing expected entanglement risk reduction to expected fishery losses, relative to status quo. All analyses were repeated with a focus on the risk posed by, and revenue expected for, small vessels (less than 12 m) only, a group that represents a substantial portion of the crab fishing fleet. Because of their smaller size, these vessels have more limited long-distance mobility and smaller on-board storage, and yet are often more active in the crab fishery later in the fishing season. Therefore, small vessels likely have a reduced capacity to adapt to significant management changes [30].

A key consideration in implementing the scenarios is the redistribution of displaced fishing activity (and associated landings and revenues) through time and across space for scenarios in which the season opening is delayed and/or some areas are subject to spring-season closures/restrictions but others are not. We evaluated several methods of redistribution, designed to consider different possible fisher behaviours, and chose to lag fishing activity by the length of any delays in season opening and to redistribute fishing activity from areas subject to spring-season actions into areas that remained unrestricted (electronic supplementary material). That is, we assumed that fishing activity in areas affected by restrictions would redistribute to areas that were less constrained, as opposed to ceasing completely (electronic supplementary material, figure S1).

## 3. Results

Within Dungeness crab fishing grounds, whale distribution models predicted more than a doubling in blue whale probability of occurrence (figure 1*a*) and humpback whale density (figure 1*b*) during the heatwave compared to pre-heatwave, with a return to lower predicted occurrence and density post-heatwave. During the heatwave, the models predicted higher blue whale probability of occurrence off Point Arena and Monterey Bay and higher humpback whale densities throughout California, but especially from Big Sur to north of San Francisco Bay (table 1). Previously undeveloped, remotely sensed VMS data from greater than 16 000 fishing trips show that, in comparison to the whale distributions, total Dungeness crab fishing activity in California increased but did not change as substantially as whale occurrence across the three periods. Fishing activity intensified off Monterey Bay and north of Point Arena during and after the heatwave, especially offshore (figure 1*c*).

**Table 1.** Summary of scenarios evaluated to consider potential impacts on the Dungeness crab fishery, and associated risk to blue and humpback whales. Early season (November–December) actions were limited to delaying the season opening until 15 December, at either of two spatial extents (statewide or central management area only). Spring-season (April–July) actions began 1 April and included early closures (100% effort reductions), 50% effort reductions and depth restrictions (closures greater than 30 fathoms) applied statewide, within the central management area only, or within blue and humpback whale biologically important areas (BIAs). See Methods and electronic supplementary material for details.

| scenario | delayed season openings | spring-season restrictions |
| --- | --- | --- |
| 1 | statewide | normal |
| 2 | central | normal |
| 3 | normal | statewide |
| 4 | normal | central |
| 5 | normal | BIAs |
| 6 | statewide | statewide |
| 7 | statewide | central |
| 8 | central | statewide |
| 9 | central | central |
| 10 | statewide | BIAs |
| 11 | central | BIAs |
| 12 | normal | 50% effort |
| 13 | normal | central 50% effort |
| 14 | normal | depth |
| 15 | normal | central depth |
| 16 | normal | depth + 50% effort |
| 17 | normal | central depth + 50% effort |
| 18 | normal | central 50% effort + depth |

An index of entanglement risk that combines the dynamic whale distribution models and data on nearly 400 000 vessel fishing locations shows that risk rose in 2015, peaked in 2016, and was coincident with ninefold higher reporting of entangled whales. Entanglements of humpback whales were responsible for the majority of this increase, though importantly blue whales were reported as entangled for the first time in the four-decade time series (figure 2a). During the heatwave period, we found that the predicted overlap between whales and fishing activity more than doubled for blue whales (figure 2b) and tripled for humpback whales (figure 2c) beginning in the 2015 crab season, compared with the previous five fishing seasons. Estimated entanglement risk to blue and humpback whales remained elevated through the 2018 fishing season, but declined in the 2019 season. By contrast to the patterns of entanglement reports and risk, revenue to the fishery did not exhibit a pronounced change in 2015, but doubled between 2016 and 2017 (figure 2d).

Strikingly, we found that none of the management scenarios we evaluated could completely mitigate the environmentally driven increase in entanglement risk during the heatwave. Under all scenarios, entanglement risk was on average fivefold higher for blue whales, and threefold higher for humpback whales, during this period compared with the others (figure 3a,b). In addition to this strong heatwave signal, there were also clear differences in entanglement risk between management scenarios during all three periods. For both blue and humpback whales, the scenario that shortened the fishing season the most (a delayed opening and a spring closure) produced the greatest reduction in entanglement risk to whales (by greater than 50% on average; figure 3a,b). However, due to considerable interannual variability, this scenario reduced entanglement risk from 10 to 100% in different years. By contrast, the other scenarios generally produced reductions in entanglement risk of 10–30% for both whale species. Importantly, since depth restrictions compressed fishing activity (figure 1c) toward the coast without reducing overall effort during the heatwave, these measures counterproductively increased entanglement risk for both whale species (figure 3a,b). Application of these scenarios within central California alone diminished the magnitude of risk reduction, but did not alter the rank order of entanglement risk reduction substantially (electronic supplementary material, figure S2a,b).

All of these management scenarios led to greater variability in outcomes for the Dungeness crab fishery during the heatwave, compared to the pre-heatwave period, but did not reveal consistent differences in average expected fishery revenue (figure 3c). Not surprisingly, the scenario that shortened the fishing season the most (a delayed opening and spring closure) produced the greatest expected losses in fishery revenue, varying from 5 to 20% across years (except for 2016, in which this scenario would have led to near zero revenue; figure 3c). In any individual fishing season, all other scenarios generally caused changes in fishery revenue of less than 20%, and expected revenue losses varied by approximately 10% between scenarios and within

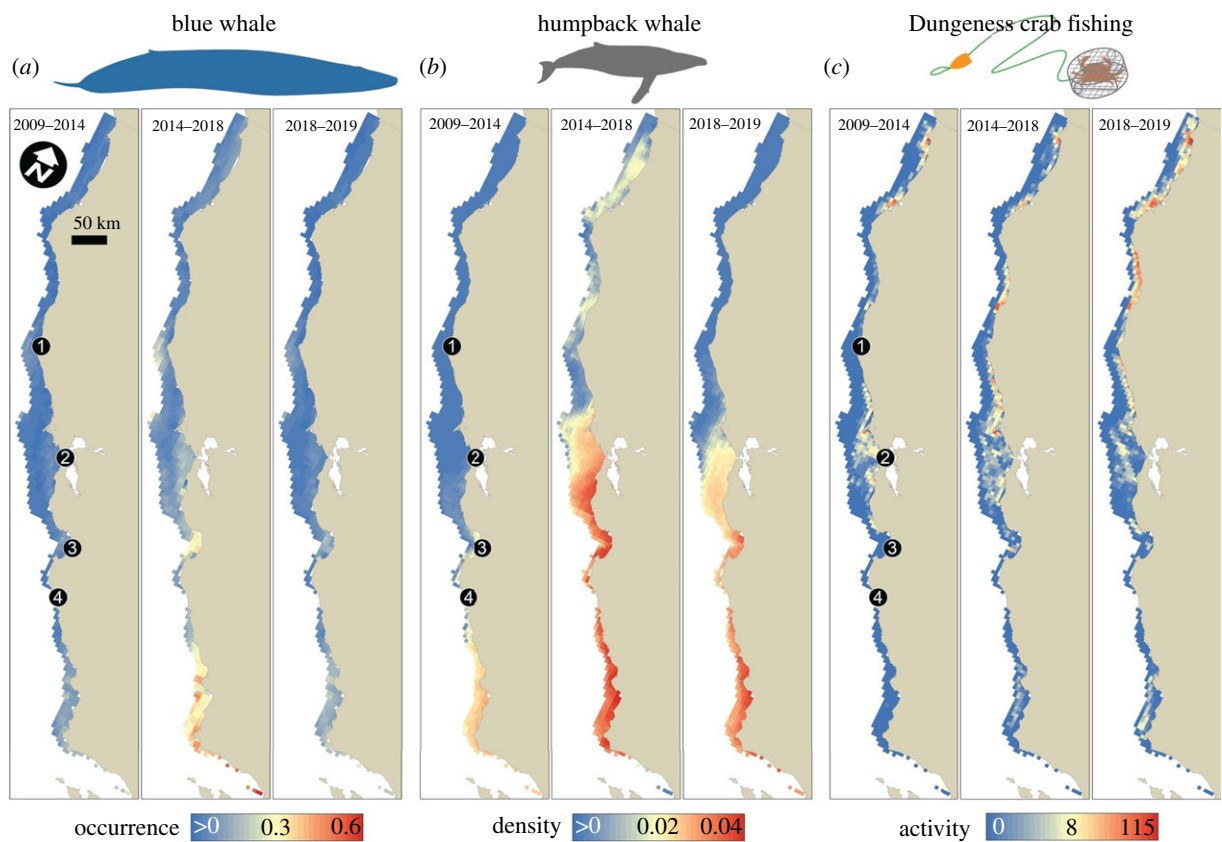

**Figure 1.** Comparison of whale and Dungeness crab fishery distributions before, during and after the Northeast Pacific marine heatwave. Predicted (*a*) blue whale probability of occurrence, (*b*) humpback whale densities and (*c*) California Dungeness crab fishing activity during three time periods representing before (2009–2014), during (2014–2018) and after (2018–2019) the marine heatwave. Values represent medians for each time period within California Dungeness crab fishing grounds (5 km grid cells with positive fishing activity) when the fishery was open (typically November–July). For (*c*), only fishing grounds with median values greater than zero for each time period are shown. Geographical reference points: (1) Point Arena, (2) San Francisco Bay, (3) Monterey Bay and (4) Big Sur. The time periods reflect 5, 4 and 1 crab fishing years, respectively; see electronic supplementary material. (Online version in colour.)

each year. Application of the scenarios in central California, but not elsewhere in state waters, reduced expected revenue losses, but the rank order of fishery revenue reduction among scenarios remained similar (electronic supplementary material, figure S2*c*). We also investigated the potential impacts of these scenarios on smaller vessels within the Dungeness crab fleet. While accounting for only approximately 11% of all fishery revenue from 2009 to 2019, smaller vessels represented approximately 45% of all vessels but generated one-third of entanglement risk compared to the full fleet. Further, we did not find qualitative differences in rank performance of scenarios for small vessels (electronic supplementary material, figures S3 and S4).

Together, these results imply that trade-offs between entanglement risk and fishery revenue became more severe during the heatwave (figure 4). The anticipated conservation benefits of management interventions were detectable and variable, with some reducing anticipated whale entanglement risk twice as much during the heatwave than before it (figures 3 and 4). However, the expected costs to the fishery escalated disproportionately, by as much as fourfold (figures 3 and 4). For scenarios involving statewide spring closures in particular, expected losses during the heatwave averaged 35% (relative to status quo) compared to 10% pre-heatwave. By contrast, on average this subset of scenarios reduced entanglement risk to blue whales by approximately 50% pre-heatwave but by approximately 65% during the heatwave, and to humpback whales by approximately 30% pre-

heatwave, but by approximately 45% during the heatwave (figures 3*a,b* and 4). Therefore, management interventions were generally less cost-effective during the heatwave compared with the pre- and post-heatwave periods, though their rank cost-effectiveness did not vary much (electronic supplementary material, figures S5 and S6). During the heatwave, there was no single management strategy that reduced entanglement risk substantially and left minimal impact on fishery revenue. Across all time periods, delaying the crab season opening at the spatial scale of the entire state and ending the season early in spring at the spatial scale of Central California was relatively cost-effective while also reducing entanglement risk by 30–50% (figure 3*a,b*; electronic supplementary material, figures S5 and S6). During the 2018–2019 period, simulated management scenarios produced little change from the status quo compared to the pre-heatwave and heatwave periods (figures 3 and 4; electronic supplementary material, figure S2).

## 4. Discussion

While gradual environmental change has subtle and lasting effects that allow for adaptation, the role of extreme events in driving coupled social–ecological dynamics is acute and difficult to buffer against [1,2,44]. Across systems there is a gap in understanding of the dynamic nature of trade-offs and how management interventions can and cannot

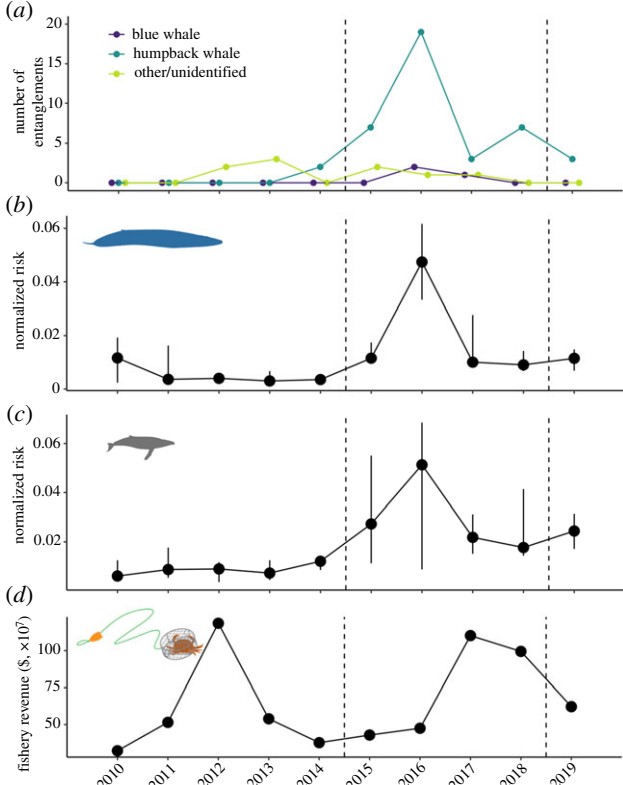

**Figure 2.** Time series of entanglement reports, risk to blue and humpback whales and California Dungeness crab fishery revenue. Changes over time (2009–2019) in (*a*) the number of confirmed whale entanglements reported in California commercial Dungeness crab gear along the US West Coast; median monthly risk (±25%) to (*b*) blue and (*c*) humpback whales from the California Dungeness crab fishery, measured as spatial overlap (see electronic supplementary material) and (*d*) revenue to the California Dungeness crab fishery (VMS-tracked vessels only). Years refer to Dungeness crab fishing seasons and dashed lines distinguish the three time periods compared in the study, which correspond to before, during and after the marine heatwave: 2009–2014, 2014–2018 and 2018–2019. In (*a*), the 'other/unidentified' group includes grey, killer and unidentified whales. (Online version in colour.)

influence them [45]. On the one hand, it is possible that climate extremes may lead to shifts in the distribution of species and human activities that could ameliorate conflict and perhaps even foster opportunities for new refugia for species and access for people [10]. On the other hand, if climate extremes stress both the human system and the wildlife system (e.g. by increasing overlap; [7]), the outer bound of points reflecting all possible combinations of social and ecological outcomes on a trade-off surface (i.e. the efficiency frontier; [43]) may move inward, reducing the opportunity for substantive win–wins. In this case study of whales and the California Dungeness crab fishery, we brought new data and models to bear on the question of which management strategies can best reduce conflict in the case of extreme warming and consequent distributional shifts. We demonstrated that the marine heatwave moved the efficiency frontier in trade-offs between fishery and conservation goals toward the origin (compare figure 4*a* versus 4*c* and 4*b* versus 4*d*). Therefore, there was no silver-bullet strategy to reduce the severity of dynamic social–ecological trade-offs during this climate extreme. This situation and others demand innovative alternative solutions to resolve

human–wildlife conflict satisfactorily and sustainably during extreme climate events.

Stopping or reducing entanglements (figure 2*a*) amidst a perfect storm of environmental change is not simple. California's RAMP and the State of California have developed new regulations representing a nascent form of dynamic and adaptive ocean management to address entanglement risk associated with the Dungeness crab fishery.[3] Indeed, these regulations resulted in the implementation of a statewide fishery closure in mid-April of 2019 (causing a smaller and more uniform influence of the alternative management scenarios we simulated in our analysis post-heatwave; figures 3 and 4; electronic supplementary material, figure S2). The ideas underpinning the RAMP and current regulations are qualitatively similar to the scenarios evaluated in this analysis, but had not previously been informed directly by a formal, quantitative risk assessment (figures 1–3, electronic supplementary material, figures S2–S4) or trade-off analysis (figure 4). Here we found that on the one hand, prior to an extreme warming event in 2014, whales were less common close to the California coast (figure 1*a*,*b*). During this pre-heatwave period, management interventions could produce near win–wins by reducing entanglement risk to blue and humpback whales at relatively low cost to the Dungeness crab fishery (as a whole fleet (figure 4) and to small vessels alone (electronic supplementary material, figure S4)), which operated most intensively close to the coast (figure 1*c*). From a whale conservation perspective alone (figure 3*a*,*b*), this finding implies that consideration of costs may have relatively little influence on selection of a management strategy to maximize entanglement risk reduction under normal ocean conditions (electronic supplementary material, figures S5 and S6). If so, the combination of a delayed season opening and an early spring closure to the fishery may be favoured in conditions like those characterizing the pre-heatwave period. However, stakeholder preferences and perceptions around feasibility, equity and evidence that improvements will be sustained across a range of ocean conditions, and factors external to our analysis (such as enforcement), will likely play a substantial role as well [46,47].

On the other hand, our analysis also clearly demonstrated that the 2014–2016 Northeast Pacific marine heatwave tipped the scales in this human–wildlife conflict, so that many of the adaptive management scenarios created more win–lose situations that successfully reduced risk to whales at great cost to the fishery (as a whole fleet (figure 4; electronic supplementary material, figures S5 and S6) and to small vessels alone (electronic supplementary material, figure S4)). Assuming that society places equal weight on changes in risk to whales and revenue to the fishery, the expected costs to the Dungeness crab fishery increased disproportionately to the expected benefits to whales (figure 4; electronic supplementary material, figures S5 and S6). The increased occurrence of whales, along with their shoreward distribution shift, during the heatwave are partly responsible for this change (figure 1*a*,*b*). The greater expected losses to the fishery during the heatwave period are in part a result of fishery management decisions addressing human and crab population health. Heatwave-associated harmful algal bloom events in 2015 and 2016 [32] and northern California product-quality concerns in 2017 delayed the opening of the fishery in three consecutive years, and the entanglement situation essentially aggravated this already difficult situation for fishery participants.[4] Perhaps as a

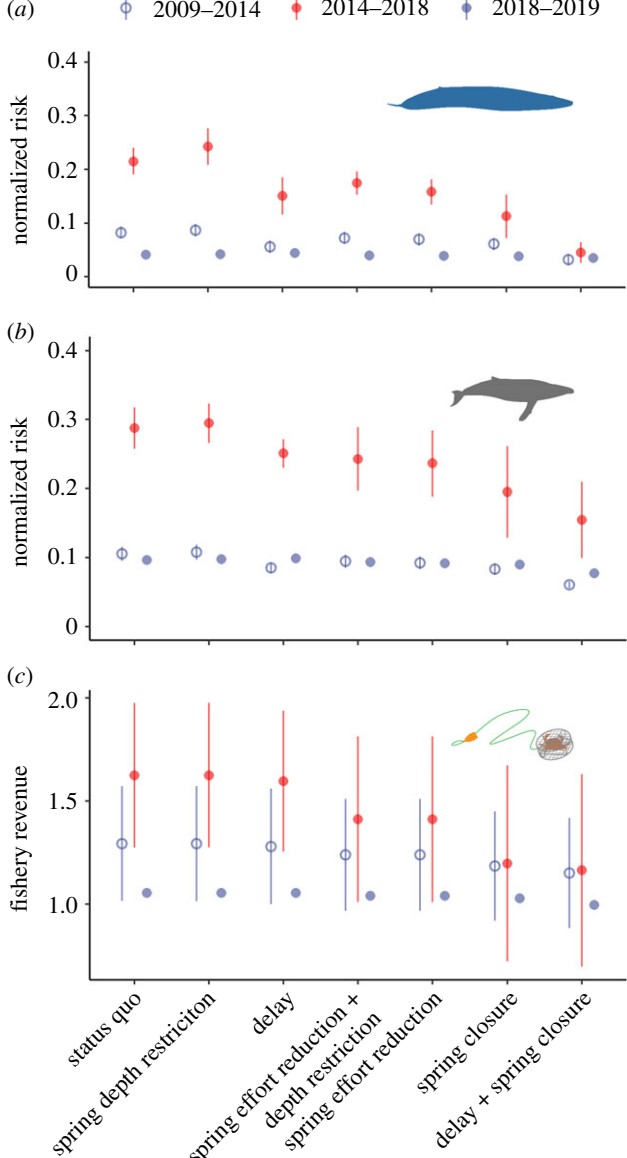

**Figure 3.** Comparison of entanglement risk and fishery revenue across alternative management scenarios and time periods. Expected risk to (*a*) blue whales and (*b*) humpback whales and (*c*) revenue to the Dungeness crab fishery under a range of status quo and alternative management scenarios affecting the entire state of California, during each of three time periods representing before (2009–2014), during (2014–2018) and after (2018–2019) the Northeast Pacific marine heatwave. In (*a*) and (*b*), each point reflects the mean (±1 s.e.) cumulative annual risk across crab years while in (*c*) each point reflects the mean (±1 s.e.) cumulative revenue across crab years. Note that there are no error bars for 2018–2019 because it represents only a single crab year. (Online version in colour.)

consequence of these contracted seasons, we observed an overall increase in fishing activity when it was open during the heatwave, especially in the central California region (figure 1*c*). However, the parallel climate-related impacts to the crab fishery also likely unintentionally reduced risk to whales by keeping fishing gear out of the water in the late autumn of those years. Furthermore, in absolute terms fishery revenue remained quite high during 2016 and 2017, due in part to crab population cycles [48] and intense effort during the months the fishery was open (figure 1*c*).

During extreme climate events like the one considered here, a subset of management actions may be more cost-effective than others and proactive measures may effectively mitigate some of the social and ecological impacts. However, it is also likely that creative solutions will be needed to achieve outcomes that are both socially and ecologically sustainable by avoiding large marginal costs without comparable gains [49]. These solutions can only be developed with a clear-headed idea of the relative weight of societal values placed upon conservation (whale recoveries) and extractive use (sustaining the fishery) goals [43], choices that can have strong influence on the location of the efficiency frontier on trade-off surfaces. As in other contexts, placing explicit values, trade-offs and cost-effectiveness at the centre of considerations of management and policy alternatives—rather than relying exclusively on a benefits-only framework—will lead decision makers to different conclusions about best practices [50,51].

One potential approach to tackling difficult trade-offs is to combine fine-scale spatial and temporal management measures with incentives designed to encourage conservation and mitigate economic loss. In the situation we consider here on the US west coast, finer-spatial scale and temporally targeted management measures appeared to achieve most of the risk reduction benefits for both blue and humpback whales at the least cost to the fishery (state-wide delay and central California spring closure; figure 4; electronic supplementary material, figures S5 and S6). This finding adds to a burgeoning literature encouraging avoidance of spatially and temporally dynamic bycatch hotspots for species of conservation concern using best available scientific information [52]. However, even for this subset of strategies, the perceived and realized costs to the fishery may be too high to be widely accepted and the expected conservation benefits too low to be considered sufficient. Additionally, time–area closures can lead to low-cost displacement of fishing effort into areas that are used by bycatch species and remain open to fishing (e.g. the spring depth restrictions and closures in BIAs examined here). The potential for these counterproductive outcomes emphasizes the need for continued progress toward other technological and policy solutions like gear innovation [53] and incentive-based measures (e.g. Payments for Ecosystem Services [54]) that are inherently more responsive to changing ecological and social conditions [55]. Technological, incentive-based measures may impose the precaution required to moderate human activity in smaller areas and for shorter periods of time on short notice [35,36].

Given the nuance and potential for win–lose outcomes, we suggest that several advances are needed to achieve cost-effective and sustainable solutions to human–wildlife conflicts under climate extremes. First, tools and information that anticipate conditions with advance warning or provide real-time evaluations of current conditions will be invaluable [36]. Second, and related, dynamic optimization algorithms, well-established in the conservation planning literature [51] and tuned to current environmental conditions, species distributions and human use patterns [56], will allow comparison of feasible management options with idealized outcomes under climate extremes. Third, we encourage future work to determine how extreme events and incentive-based management measures will change how people make decisions about where they use the ocean (e.g. fishing grounds).

Both gradual and event-driven ecosystem change in other natural resource management contexts will result in more novel risks that require similarly proactive and adaptive

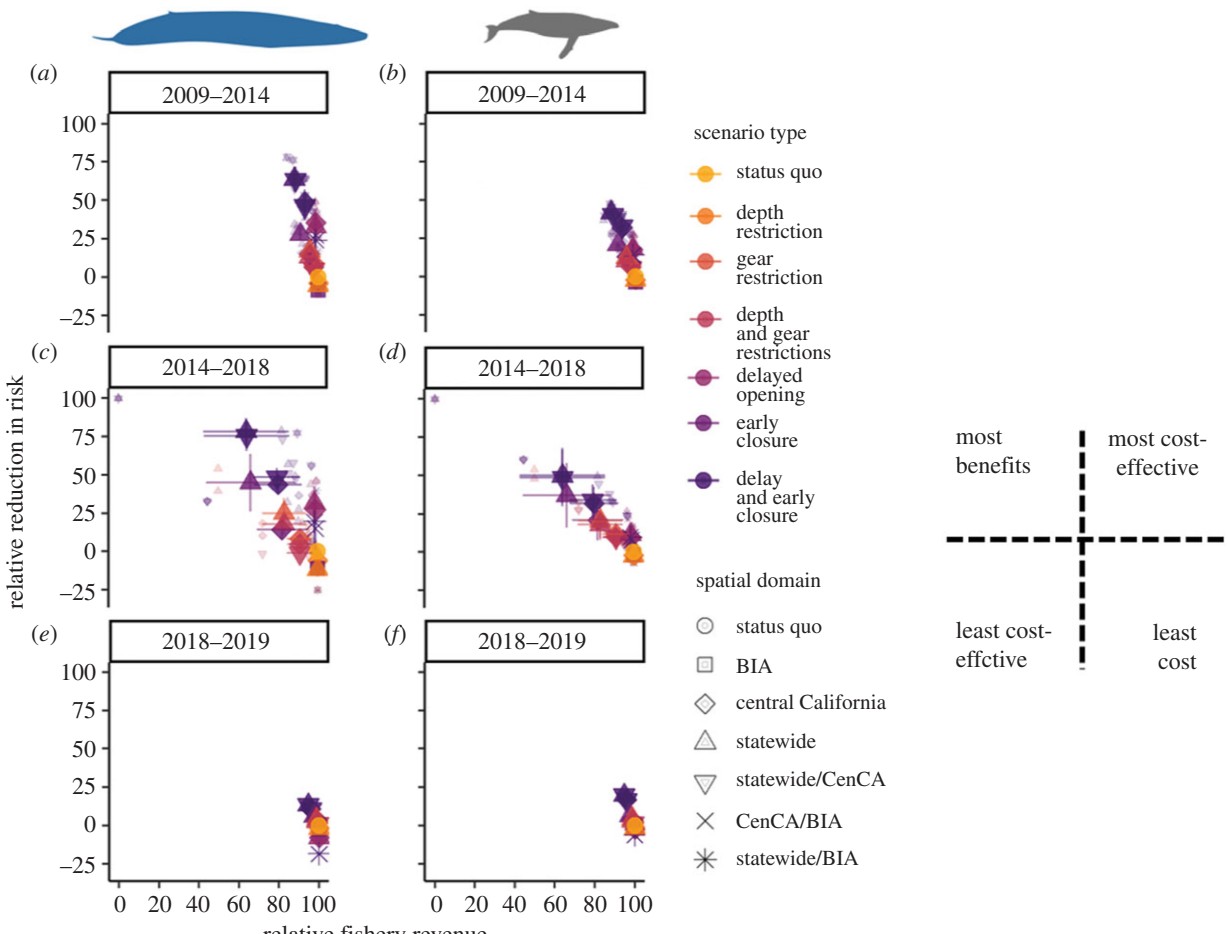

**Figure 4.** Trade-offs between risk of whale entanglement and California Dungeness crab revenue across alternative management scenarios and time periods. Plots indicate the expected reduction in (*a,c,e*) risk to blue whales or (*b,d,f*) to humpback whales in comparison to expected revenue to the California Dungeness crab fishery, relative to status quo, under a range of alternative management scenarios, during each of three time periods representing before (2009–2014; *a,b*), during (2014–2018; *c,d*) and after (2018–2019; *e,f*) the Northeast Pacific marine heatwave. Larger points and error bars represent the median ± 1 s.e. for each scenario across years, smaller points represent values for individual years. Circular points in the lower right of each panel represent the status quo (100% fishery revenue and 0% risk reduction in each year). The inset figure on the right provides a guide for interpreting where different points representing the alternative scenarios fall within the trade-off space, such that those in the upper right (lower left) are most (least) cost-effective while those in the upper left (lower right) provide the most (least) benefits at greatest (lowest) cost. (Online version in colour.)

management structures to achieve sustainable use [38]. For example, forage fish collapses are frequently driven by environmental extremes, yet status quo fishing effort often amplifies these effects [57]. Along the western Atlantic coast, an environmentally driven spike in both entanglements and mortality of critically endangered North Atlantic right whales has reversed previously successful recovery efforts [21]. Similarly, unforeseen conflicts in other regions may arise due to climate variability and the development of offshore ecosystem services—such as renewable energy and aquaculture— that are difficult to modify once in place [58,59]. Though it is possible that climate extremes could reduce conflict between humans and wildlife by, for example, reducing the overlap between species and human activities, the ever-increasing footprint of human activities on land and at sea suggest that this possibility will become increasingly unlikely [7].

While accurately predicting the consequences of any specific management tactic in an uncertain future will always be riddled with problems, decision theory holds that it is possible to rank expected outcomes from alternative management actions with greater certainty [60]. The strategic use of trade-off analysis to evaluate, rank and choose between

proactive, and potentially dynamic, ocean management scenarios offers a clear path forward. Participatory processes, partnerships and cooperative efforts across institutions, such as California's RAMP, are promising avenues to ingest such information and use it to handle these issues through adaptive management and innovative policy development [17,59]. Though no panacea [61], direct cooperation and polycentric governance can encourage movement to more optimal approaches by embracing complexities in institutional arrangements and incentives [62]. Such cooperation, along with technological, data and analytical advances, can help decision makers become more agile as climate extremes and climate change create new tensions between conservation and sustainable resource management goals.

Data accessibility. All entanglement report data are available at https:// oceanview.pfeg.noaa.gov/whale_indices/. All outputs from the blue whale habitat suitability model are available at https://coastwatch. pfeg.noaa.gov/projects/whalewatch2/whalewatch2_map.html. Humpback whale model outputs available upon request to K.A.F. Confidential vessel-level landings, registration and vessel monitoring system data may be acquired by direct request from the California Department of Fish and Wildlife and the US National Marine Fisheries Service Office of Law Enforcement, subject to a non-disclosure

agreement. Aggregated data used to evaluate risk and revenue associated with specific management scenarios and all associated R code are publicly available on Github (https://github.com/jameals/raimbow).

Authors' contributions. J.F.S.: conceptualization, formal analysis, funding acquisition, investigation, methodology, project administration, resources, software, supervision, validation, visualization, writing—original draft, writing—review and editing; M.C.F.: formal analysis, funding acquisition, investigation, methodology, software, writing—review and editing; O.L.: formal analysis, investigation, methodology, software, visualization, writing—review and editing; S.M.W.: formal analysis, investigation, methodology, software, writing—review and editing; D.L.: conceptualization, funding acquisition, writing—review and editing; J.R.: methodology, validation, writing—review and editing; L.E.S.: investigation, methodology, writing—review and editing; B.E.F.: Investigation, methodology, visualization, writing—review and editing; B.A.: funding acquisition, investigation, methodology, writing—review and editing; K.A.F.: methodology, resources, validation, writing—review and editing; E.L.H.: funding acquisition, methodology, writing—review and editing. All authors gave final approval for publication and agreed to be held accountable for the work performed therein.

Competing interests. We declare we have no competing interests.

Funding. We gratefully acknowledge support from the US National Marine Fisheries Service Protected Resources Division, Western Regional Office, Office of Law Enforcement and California Current Integrated Ecosystem Assessment programme. This material is based upon work supported by the National Science Foundation Graduate Research Fellowship Programme under grant no. DGE-1762114 (M.C.F.). Benioff Ocean Initiative supported the development of the blue whale models.

Acknowledgements. This manuscript benefited from discussions with the California Dungeness Crab fishing gear working group, C.D.F.W., D. Bradley, C. Free, L. Bellquist, J. Wilson, K. Kauer, A. Jackson, C. Shuman, J. Santora and D Holland, and reviews by C.J. Harvey, M. Savoca, K. Arkema, M. Ruckelshaus, S.K. Moore and A.C. Stier.

## Endnotes

[1]See Habitat Compression Index at the California Current Integrated Ecosystem Assessment website. https://oceanview.pfeg.noaa.gov/dashboard/.

[2]14 CCR § 132.8; WAC 220-340-480; OAR 635-005-0405 and 635-005-0460.

[3]In 2019 and 2020, CDFW implemented these new measures in an effort to reduce entanglement risks. https://wildlife.ca.gov/conservation/marine/whale-safe-fisheries#55999897-risk-assessment.

[4]The history of delayed openings is available here under regulations and corresponding month/year combinations. https://cdfwmarine.wordpress.com/.

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
