## [Peer Review File · Proceedings of the Royal Society B: Biological Sciences]

Review History

RSPB-2021-1607.R0 (Original submission)

Review form: Reviewer 1

Recommendation

Major revision is needed (please make suggestions in comments)

Scientific importance: Is the manuscript an original and important contribution to its field?

Good

General interest: Is the paper of sufficient general interest?

Good

Quality of the paper: Is the overall quality of the paper suitable?

Acceptable

Is the length of the paper justified?

Yes

Should the paper be seen by a specialist statistical reviewer?

No

Do you have any concerns about statistical analyses in this paper? If so, please specify them explicitly in your report.

No

It is a condition of publication that authors make their supporting data, code and materials available - either as supplementary material or hosted in an external repository. Please rate, if applicable, the supporting data on the following criteria.

Is it accessible?

No

Is it clear?

N/A

Is it adequate?

No

Do you have any ethical concerns with this paper?

No

Comments to the Author

This manuscript considers how the effects of extreme climatic events, specifically marine heatwaves, can influence management that impacts whale entanglement and the revenue obtained from the California Dungeness crab fishery. This is a timely topic for consideration given the increasing frequency, magnitude, and duration of marine heatwaves being driven by global climate change. While the topic is of interest, the current manuscript could be revised and restructured to enhance the clarity of the message delivered to readers. Suggestions on how to do this follow:

Major comments

Abstract and Introduction – Page 5, paragraph 2: The need for trade-offs between whale conservation goals and the sustainability of the California Dungeness crab fishery are described toward the end of the Introduction. It would be useful for the reader to have this interaction introduced earlier, such that the importance of the crab fishery – and link to whale conservation – is recognised and its significance understood.

It would also be good to specify this is the fishery of interest in the Abstract of the paper.

Methods: It isn't clear to me how the different "scenario types" or "spatial domains" considered were selected, or what, exactly, they represent (e.g. what depths are covered in the depth restriction? Which gear types addressed? How much is opening delayed by? How much early is closure?). Some, if not all, of this information appears to be in Table 1 and the Supplementary Material, but could be of benefit if included in the main text.

Results: Throughout the Results there are lines that should be moved to other sections (either the Introduction, Methods, or the Discussion), for example: "Despite its enormous value, until now there has been no rigorous examination of the spatial dynamics of the crab fishery using data collected independently and at fine spatial scales"; "To provide finer-grained insight into these linked social and ecological dynamics, we developed an index of entanglement risk that combines the dynamic whale distribution models and data on nearly 400,000 vessel fishing locations.". Similarly, the final paragraph of the Results reads like a summary that should be in the Discussion.

There also should not be statements that require references in the Results as they belong in the Discussion – this section should focus on what has been found in the current study and context provided elsewhere in the manuscript.

The figures and figure panels should be in the same order in text as they are presented in the figures.

Discussion: The Discussion covers the context in which the study occurs (some of which may be better in the Introduction – e.g. the beginning of paragraph 3 on page 9 discussing the regulatory history), but has very little exploration of the results obtained in this specific study. I feel that this is an area that could be expanded upon (particularly given that some of what could be in the Discussion is currently in the Results). For example, of the four figures included in the main text, only Figure 4 was mentioned in the Discussion.

Moreover, the Introduction concludes with the statement that the authors: “(1) evaluate social consequences of this human-wildlife conflict, (2) quantify the spatial and temporal dynamics behind them, and (3) explore the potential for dynamic management strategies to mitigate tradeoffs between whale conservation goals and the sustainability of the California Dungeness crab fishery before, during, and after this period of social-ecological squeeze”. From my reading point 3 is most clearly covered in the Discussion, but the others are not to the same extent. Statements made in the Discussion should also be more consistently supported with references, possibly to the grey literature, particularly for pages 9-10, paragraphs 3-2.

Minor comments

Abstract: It would be good to specify the marine heatwave that is discussed in the manuscript here.

It would be useful to indicate which of the groups are the “losers” in the win-lose outcomes.

Page 4, paragraph 1: Add a reference for the definition of heatwaves.

Page 4, paragraph 2: Provide a reference and definition of marine heatwaves specifically – a commonly used definition can be found in:

Hobday, A. J., Alexander, L. V., Perkins, S. E., Smale, D. A., Straub, S. C., Oliver, E. C. J., Benthuyesen, J. A., Burrows, M. T., Donat, M. G., Feng, M., Holbrook, N. J., Moore, P. J., Scannell, H. A., Sen Gupta, A., & Wernberg, T. (2016). A hierarchical approach to defining marine heatwaves. *Progress in Oceanography*, 141, 227–238.

<https://doi.org/10.1016/j.pcean.2015.12.014>

Or, alternatively:

Hobday, A. J., Oliver, E., Sen Gupta, A., Benthuyesen, J., Burrows, M., Donat, M., Holbrook, N., Moore, P., Thomsen, M., Wernberg, T., & Smale, D. (2018). Categorizing and Naming Marine Heatwaves. *Oceanography*, 31(2). <https://doi.org/10.5670/oceanog.2018.205>

Oliver, E. C. J., Donat, M. G., Burrows, M. T., Moore, P. J., Smale, D. A., Alexander, L. V., Benthuyesen, J. A., Feng, M., Sen Gupta, A., Hobday, A. J., Holbrook, N. J., Perkins-Kirkpatrick, S. E., Scannell, H. A., Straub, S. C., & Wernberg, T. (2018). Longer and more frequent marine heatwaves over the past century. *Nature Communications*, 9(1), 1324.

<https://doi.org/10.1038/s41467-018-03732-9>

Page 4, paragraph 2: It is noted that in “many regions around the world large whales have experienced changes in the timing and pathways of migration and occurrence on feeding grounds”. It would be good to indicate if these changes are driven by heatwaves specifically, or due to other changes (e.g. ocean warming, acidification, pollution, short-term upwelling, etc), and to provide supporting references.

Pages 4-5: It is noted that “changing environmental conditions, higher whale abundance and distributional shifts, and an increasingly crowded ocean” contribute to challenging conservation of whales. A case study is then discussed that is intended to highlight these points, but to me it isn’t clear how these three aspects are exemplified (e.g. there is no discussion of how whale shifts have contributed).

Page 5, paragraph 2: In the win-lose outcome, is it the protected species or fisheries that are winning/losing?

Review form: Reviewer 2

Recommendation

Accept with minor revision (please list in comments)

Scientific importance: Is the manuscript an original and important contribution to its field?

Good

General interest: Is the paper of sufficient general interest?

Good

Quality of the paper: Is the overall quality of the paper suitable?

Excellent

Is the length of the paper justified?

Yes

Should the paper be seen by a specialist statistical reviewer?

No

Do you have any concerns about statistical analyses in this paper? If so, please specify them explicitly in your report.

No

It is a condition of publication that authors make their supporting data, code and materials available - either as supplementary material or hosted in an external repository. Please rate, if applicable, the supporting data on the following criteria.

Is it accessible?

Yes

Is it clear?

Yes

Is it adequate?

Yes

Do you have any ethical concerns with this paper?

No

Comments to the Author

This is a rich, and well-written study touching on at least two important topics: first, how a specific marine heat wave in the western U.S. affected a specific – albeit important – human-wildlife conflict (Dungeness crab fishing and whale entanglement); and second, how climate extremes might affect human wildlife-conflict and the availability of win-wins (or near win-wins) generally. It should be published with some minor-to-medium revisions.

I think the first of these topics – the crab-whale conflict – is covered very well. The authors compellingly show and explain how the heat wave changed this specific conflict by altering

timing and distributions of fishing effort and whales.

However, I think the authors should dig into the second topic a bit more in the discussion and overall framing of the article. Specifically, it would be good to give the reader a more general sense, conceptually, of how climate extremes should be expected to affect human-wildlife conflicts. For instance, I can think of two general types of effects that climate extremes could have, one of which I would always expect to exacerbate the conflict, as in this particular case, but the other where you might see variation on a case-by-case basis.

Type 1: the extreme stresses both the human and wildlife system. Adding stress to both objectives moves the efficiency frontier towards the origin, which should make win-wins harder to find. To what extent does this mechanism apply here?

Type 2: the extreme moves species and fishing effort around in space and time. This seems like it shouldn't always make the conflict worse. In the whale-crab case it did (by making them coincide more), but I don't see why they should always do that (but if the authors think there is a reason, that would be interesting and worth stating).

The authors don't necessarily need to use these classifications, but they should describe, at a conceptual level, how generalizable they expect climate extremes exacerbating human-wildlife conflict to be in other systems, and why that is.

Minor comments:

- Figure 3 caption: please remove tacked changes.
- Is it possible to include points representing the status quo in Fig. 4?
- Figure S2: why are the changes so much smaller and more uniform after the heat wave (2018-2019)? Is that related to the timing conflict with the migration not subsiding when the heatwave ended, as mentioned on page 8?

Decision letter (RSPB-2021-1607.R0)

16-Sep-2021

Dear Dr Samhuri

I am pleased to inform you that your manuscript RSPB-2021-1607 entitled "Marine heatwave challenges solutions to human-wildlife conflict" has been accepted for publication in Proceedings B.

The referee(s) have recommended publication, but also suggest some minor revisions to your manuscript. Therefore, I invite you to respond to the referee(s)' comments and revise your manuscript. Because the schedule for publication is very tight, it is a condition of publication that you submit the revised version of your manuscript within 7 days. If you do not think you will be able to meet this date please let us know.

It is a condition of publication that data supporting your paper are made available either in the electronic supplementary material or through an appropriate repository. Please see our Data Sharing Policies <https://royalsociety.org/journals/authors/author-guidelines/#data>.

[http://datadryad.org/submit?journalID=RSPB&manu=\(Document not available\)](http://datadryad.org/submit?journalID=RSPB&manu=(Document not available)) which will take you to your unique entry in the Dryad repository. If you have already submitted your data to dryad you can make any necessary revisions to your dataset by following the above link.

Please see <https://royalsociety.org/journals/ethics-policies/data-sharing-mining/> for more details.

Sincerely,
Dr Locke Rowe
mailto: proceedingsb@royalsociety.org

Associate Editor
Board Member: 1

Comments to Author:

This manuscript presents interesting results relating to the impacts of extreme climate events using a social and ecological systems lens. The authors critically evaluate a record marine heatwave and related conflict between conservation goals and human use through a case study with a unique emphasis on innovative management and policy interventions. This manuscript is clearly written, and the results are novel. The manuscript has been reviewed by two reviewers. Both found aspects of this paper interesting, but Reviewer 2 pointed out a number of ways in which the manuscript can improve. In particular, Reviewer 2 asks for further details on the methods to be provided in the main text of the manuscript. In addition, Reviewer 2 provides a comprehensive summary of several refinements to the Methods and Discussion section that would improve the manuscript. Specifically, the Discussion section needs to be revised and the findings considered in detail in the context of the study area and with reference to additional peer-reviewed literature. Further, Reviewer 1 highlights the need to expand on the text related to how climate extremes might affect human-wildlife conflicts in the Discussion section. These revisions will ultimately improve the manuscript.

Reviewer(s)' Comments to Author:

Referee: 1

Comments to the Author(s)

This manuscript considers how the effects of extreme climatic events, specifically marine heatwaves, can influence management that impacts whale entanglement and the revenue obtained from the California Dungeness crab fishery. This is a timely topic for consideration given the increasing frequency, magnitude, and duration of marine heatwaves being driven by global climate change. While the topic is of interest, the current manuscript could be revised and restructured to enhance the clarity of the message delivered to readers. Suggestions on how to do this follow:

Major comments

Abstract and Introduction – Page 5, paragraph 2: The need for trade-offs between whale conservation goals and the sustainability of the California Dungeness crab fishery are described toward the end of the Introduction. It would be useful for the reader to have this interaction introduced earlier, such that the importance of the crab fishery – and link to whale conservation – is recognised and its significance understood.

It would also be good to specify this is the fishery of interest in the Abstract of the paper.

Methods: It isn't clear to me how the different "scenario types" or "spatial domains" considered were selected, or what, exactly, they represent (e.g. what depths are covered in the depth restriction? Which gear types addressed? How much is opening delayed by? How much early is closure?). Some, if not all, of this information appears to be in Table 1 and the Supplementary Material, but could be of benefit if included in the main text.

Results: Throughout the Results there are lines that should be moved to other sections (either the Introduction, Methods, or the Discussion), for example: “Despite its enormous value, until now there has been no rigorous examination of the spatial dynamics of the crab fishery using data collected independently and at fine spatial scales”; “To provide finer-grained insight into these linked social and ecological dynamics, we developed an index of entanglement risk that combines the dynamic whale distribution models and data on nearly 400,000 vessel fishing locations.”. Similarly, the final paragraph of the Results reads like a summary that should be in the Discussion.

There also should not be statements that require references in the Results as they belong in the Discussion – this section should focus on what has been found in the current study and context provided elsewhere in the manuscript.

The figures and figure panels should be in the same order in text as they are presented in the figures.

Discussion: The Discussion covers the context in which the study occurs (some of which may be better in the Introduction – e.g. the beginning of paragraph 3 on page 9 discussing the regulatory history), but has very little exploration of the results obtained in this specific study. I feel that this is an area that could be expanded upon (particularly given that some of what could be in the Discussion is currently in the Results). For example, of the four figures included in the main text, only Figure 4 was mentioned in the Discussion.

Moreover, the Introduction concludes with the statement that the authors: “(1) evaluate social consequences of this human-wildlife conflict, (2) quantify the spatial and temporal dynamics behind them, and (3) explore the potential for dynamic management strategies to mitigate tradeoffs between whale conservation goals and the sustainability of the California Dungeness crab fishery before, during, and after this period of social-ecological squeeze”. From my reading point 3 is most clearly covered in the Discussion, but the others are not to the same extent. Statements made in the Discussion should also be more consistently supported with references, possibly to the grey literature, particularly for pages 9-10, paragraphs 3-2.

Minor comments

Abstract: It would be good to specify the marine heatwave that is discussed in the manuscript here.

It would be useful to indicate which of the groups are the “losers” in the win-lose outcomes.

Page 4, paragraph 1: Add a reference for the definition of heatwaves.

Page 4, paragraph 2: Provide a reference and definition of marine heatwaves specifically – a commonly used definition can be found in:

Hobday, A. J., Alexander, L. V., Perkins, S. E., Smale, D. A., Straub, S. C., Oliver, E. C. J., Benthuyesen, J. A., Burrows, M. T., Donat, M. G., Feng, M., Holbrook, N. J., Moore, P. J., Scannell, H. A., Sen Gupta, A., & Wernberg, T. (2016). A hierarchical approach to defining marine heatwaves. *Progress in Oceanography*, 141, 227–238.
<https://doi.org/10.1016/j.pocean.2015.12.014>

Or, alternatively:

Hobday, A. J., Oliver, E., Sen Gupta, A., Benthuyesen, J., Burrows, M., Donat, M., Holbrook, N., Moore, P., Thomsen, M., Wernberg, T., & Smale, D. (2018). Categorizing and Naming Marine Heatwaves. *Oceanography*, 31(2). <https://doi.org/10.5670/oceanog.2018.205>
 Oliver, E. C. J., Donat, M. G., Burrows, M. T., Moore, P. J., Smale, D. A., Alexander, L. V., Benthuyesen, J. A., Feng, M., Sen Gupta, A., Hobday, A. J., Holbrook, N. J., Perkins-Kirkpatrick, S. E., Scannell, H. A., Straub, S. C., & Wernberg, T. (2018). Longer and more frequent marine heatwaves over the past century. *Nature Communications*, 9(1), 1324.
<https://doi.org/10.1038/s41467-018-03732-9>

Page 4, paragraph 2: It is noted that in “many regions around the world large whales have experienced changes in the timing and pathways of migration and occurrence on feeding

grounds". It would be good to indicate if these changes are driven by heatwaves specifically, or due to other changes (e.g. ocean warming, acidification, pollution, short-term upwelling, etc), and to provide supporting references.

Pages 4-5: It is noted that "changing environmental conditions, higher whale abundance and distributional shifts, and an increasingly crowded ocean" contribute to challenging conservation of whales. A case study is then discussed that is intended to highlight these points, but to me it isn't clear how these three aspects are exemplified (e.g. there is no discussion of how whale shifts have contributed).

Page 5, paragraph 2: In the win-lose outcome, is it the protected species or fisheries that are winning/losing?

Referee: 2

Comments to the Author(s)

This is a rich, and well-written study touching on at least two important topics: first, how a specific marine heat wave in the western U.S. affected a specific – albeit important – human-wildlife conflict (Dungeness crab fishing and whale entanglement); and second, how climate extremes might affect human wildlife-conflict and the availability of win-wins (or near win-wins) generally. It should be published with some minor-to-medium revisions.

I think the first of these topics – the crab-whale conflict – is covered very well. The authors compellingly show and explain how the heat wave changed this specific conflict by altering timing and distributions of fishing effort and whales.

However, I think the authors should dig into the second topic a bit more in the discussion and overall framing of the article. Specifically, it would be good to give the reader a more general sense, conceptually, of how climate extremes should be expected to affect human-wildlife conflicts. For instance, I can think of two general types of effects that climate extremes could have, one of which I would always expect to exacerbate the conflict, as in this particular case, but the other where you might see variation on a case-by-case basis.

Type 1: the extreme stresses both the human and wildlife system. Adding stress to both objectives moves the efficiency frontier towards the origin, which should make win-wins harder to find. To what extent does this mechanism apply here?

Type 2: the extreme moves species and fishing effort around in space and time. This seems like it shouldn't always make the conflict worse. In the whale-crab case it did (by making them coincide more), but I don't see why they should always do that (but if the authors think there is a reason, that would be interesting and worth stating).

The authors don't necessarily need to use these classifications, but they should describe, at a conceptual level, how generalizable they expect climate extremes exacerbating human-wildlife conflict to be in other systems, and why that is.

Minor comments:

- Figure 3 caption: please remove tacked changes.
- Is it possible to include points representing the status quo in Fig. 4?
- Figure S2: why are the changes so much smaller and more uniform after the heat wave (2018-2019)? Is that related to the timing conflict with the migration not subsiding when the heatwave ended, as mentioned on page 8?

Author's Response to Decision Letter for (RSPB-2021-1607.R0)

See Appendix A.

RSPB-2021-1607.R1 (Revision)

Review form: Reviewer 1

Recommendation

Accept as is

Scientific importance: Is the manuscript an original and important contribution to its field?

Good

General interest: Is the paper of sufficient general interest?

Good

Quality of the paper: Is the overall quality of the paper suitable?

Good

Is the length of the paper justified?

Yes

Should the paper be seen by a specialist statistical reviewer?

No

Do you have any concerns about statistical analyses in this paper? If so, please specify them explicitly in your report.

No

It is a condition of publication that authors make their supporting data, code and materials available - either as supplementary material or hosted in an external repository. Please rate, if applicable, the supporting data on the following criteria.

Is it accessible?

No

Is it clear?

Yes

Is it adequate?

Yes

Do you have any ethical concerns with this paper?

No

Comments to the Author

In my view the authors have done a good job of addressing the comments from the Editor and Reviewers on the previous version, and I have no further comments

Decision letter (RSPB-2021-1607.R1)

28-Oct-2021

Dear Dr Samhuri

I am pleased to inform you that your manuscript entitled "Marine heatwave challenges solutions to human-wildlife conflict" has been accepted for publication in Proceedings B.

Data Accessibility section

Open Access

Paper charges

Sincerely,

Dr Locke Rowe

Appendix A

UNITED STATES DEPARTMENT OF COMMERCE
National Oceanic and Atmospheric Administration
NOAA Fisheries
Northwest Fisheries Science Center
Conservation Biology Division
2725 Montlake Blvd East
Seattle WA 98112-2097
jameal.samhour@gmail.com
Ph: 206-302-1740
Fax: 206-860-3475

23 September 2021

Dr. Locke Rowe
Editor, *Proceedings of the Royal Society B*

Dear Dr. Rowe,

Please find attached our revised manuscript entitled “Marine heatwave challenges solutions to human-wildlife conflict” (RSPB-2021-1607). The comments provided by the Associated Editor and Reviewers were very helpful, and we feel they have significantly improved our manuscript. We have revised it accordingly and provide detailed responses to specific comments below in *italics*.

The main criticisms of our manuscript were focused on conceptual framing and structural organization. We addressed the following major comments, including:

- Providing further details on the methods in the main text of the manuscript;
- Refining the Methods and Discussion sections as recommended by Reviewer 2, especially in order to shine greater light on our findings in the context of our study area as well as the broader literature;
- Expanding the text related to how climate extremes might affect human-wildlife conflicts in the Discussion.

We describe further details about how we approached revisions adjacent to each individual comment in the body of this letter.

Again, we very much appreciate the reviewers’ suggestions, and the changes we have made in response have improved the paper substantially. Please do not hesitate to contact me with questions regarding the revision.

Thank you in advance for your consideration.

Jameal Samhuri (on behalf of all co-authors)

Associate Editor

Board Member: 1

Comments to Author:

This manuscript presents interesting results relating to the impacts of extreme climate events using a social and ecological systems lens. The authors critically evaluate a record marine heatwave and related conflict between conservation goals and human use through a case study with a unique emphasis on innovative management and policy interventions. This manuscript is clearly written, and the results are novel. The manuscript has been reviewed by two reviewers. Both found aspects of this paper interesting, but Reviewer 2 pointed out a number of ways in which the manuscript can improve. In particular, Reviewer 2 asks for further details on the methods to be provided in the main text of the manuscript. In addition, Reviewer 2 provides a comprehensive summary of several refinements to the Methods and Discussion section that would improve the manuscript. Specifically, the Discussion section needs to be revised and the findings considered in detail in the context of the study area and with reference to additional peer-reviewed literature. Further, Reviewer 1 highlights the need to expand on the text related to how climate extremes might affect human-wildlife conflicts in the Discussion section. These revisions will ultimately improve the manuscript.

We thank the editor for their comments and suggestions. As recommended, in the revised ms we provide further details on the methods in the main text of the manuscript, and refined the Methods and Discussion sections as suggested. We paid particular attention to adding more context about the study area and highlighting the predicted shifts in whale and fishery distributions that underlie the heightened conflict we observed during the heatwave. We also followed the recommendations to zoom out and consider how climate extremes might affect human-wildlife conflicts in general in the Discussion section.

Reviewer(s)' Comments to Author:

Referee: 1

Comments to the Author(s)

This manuscript considers how the effects of extreme climatic events, specifically marine heatwaves, can influence management that impacts whale entanglement and the revenue obtained from the California Dungeness crab fishery. This is a timely topic for consideration

given the increasing frequency, magnitude, and duration of marine heatwaves being driven by global climate change. While the topic is of interest, the current manuscript could be revised and restructured to enhance the clarity of the message delivered to readers. Suggestions on how to do this follow:

Major comments

Abstract and Introduction – Page 5, paragraph 2: The need for trade-offs between whale conservation goals and the sustainability of the California Dungeness crab fishery are described toward the end of the Introduction. It would be useful for the reader to have this interaction introduced earlier, such that the importance of the crab fishery – and link to whale conservation – is recognised and its significance understood.

It would also be good to specify this is the fishery of interest in the Abstract of the paper.

In the revised ms, we specified the fishery of interest in the Abstract. While we appreciate the suggestion about motivating the specifics of our study system earlier in the ms, we note that it is introduced in the third paragraph, a fairly early position in the ms. We would prefer not to change this structure, as we feel that the revised Introduction based on the more specific comments below provides sufficient and structurally-appropriate motivation. However, if the reviewer and the Editor feel strongly about this recommendation, we propose to add this sentence to the end of the first paragraph of the Introduction: “In this paper, we detail how these general concerns apply specifically on the US west coast, as a recent climate extreme challenged both the viability of the enormously-important California Dungeness crab fishery and the recoveries of threatened and endangered large whales that can be incidentally captured in the gear used to catch the crab.”

Methods: It isn't clear to me how the different “scenario types” or “spatial domains” considered were selected, or what, exactly, they represent (e.g. what depths are covered in the depth restriction? Which gear types addressed? How much is opening delayed by? How much early is closure?). Some, if not all, of this information appears to be in Table 1 and the Supplementary Material, but could be of benefit if included in the main text.

We can understand how it was confusing that the scenario types and spatial domains were only described in the caption of Table 1 and in the Supplementary Material. We note that the scenarios were developed through multiple, informal and formal conversations with US west coast State Dungeness crab fishery managers or their advisory working groups, and/or based on scenarios recently implemented. In the revised ms, we have provided these additional details in the Methods section (lines 189-198).

Results: Throughout the Results there are lines that should be moved to other sections (either the Introduction, Methods, or the Discussion), for example:

Thank you for helping us to improve the structural organization of the ms. We have heavily revised the Introduction, Results, and Discussion to align with these suggestions. Responses to individual comments appear below.

“Despite its enormous value, until now there has been no rigorous examination of the spatial dynamics of the crab fishery using data collected independently and at fine spatial scales”;

We moved this sentence to the Methods section of the revised ms (lines 175-177).

“To provide finer-grained insight into these linked social and ecological dynamics, we developed an index of entanglement risk that combines the dynamic whale distribution models and data on nearly 400,000 vessel fishing locations.”.

We edited the beginning of this paragraph (lines 243-250) to read as follows: “An index of entanglement risk that combines the dynamic whale distribution models and data on nearly 400,000 vessel fishing locations shows that risk rose in 2015, peaked in 2016, and was coincident with nine-fold higher reporting of entangled whales. Entanglements of humpback whales were responsible for the majority of this increase, though importantly blue whales were reported as entangled for the first time in the four-decade time series (Fig. 2a). During the heatwave period, we found that the predicted overlap between whales and fishing activity more than doubled for blue whales (Fig. 2b) and tripled for humpback whales (Fig. 2c) beginning in the 2015 crab season, compared with the previous five fishing seasons.” While we understand that this revision repeats a small amount of information provided in the Methods section, it also reminds the reader what the entanglement risk index actually is, which we believe aids its interpretation.

Similarly, the final paragraph of the Results reads like a summary that should be in the Discussion.

We appreciate the reviewer’s suggestion that this paragraph could be in the Discussion. However, we prefer to include it in the Results section as it provides us the first opportunity to introduce the tradeoff (Fig. 4) and cost-effectiveness (Figs. S5-S6) figures.

There also should not be statements that require references in the Results as they belong in the Discussion – this section should focus on what has been found in the current study and context provided elsewhere in the manuscript.

We have removed statements that require references from the Results section and moved them to other sections of the manuscript.

The figures and figure panels should be in the same order in text as they are presented in the figures.

In the revised manuscript, we refer to all figures and figure panels in the same order in the text as they are presented in the figures.

Discussion: The Discussion covers the context in which the study occurs (some of which may be better in the Introduction – e.g. the beginning of paragraph 3 on page 9 discussing the regulatory history), but has very little exploration of the results obtained in this specific study. I feel that this is an area that could be expanded upon (particularly given that some of what could be in the Discussion is currently in the Results). For example, of the four figures included in the main text, only Figure 4 was mentioned in the Discussion.

We thank the reviewer for their suggestions about how to improve the Discussion. We have added references to all figures throughout the Discussion, and moved the sentence about the regulatory history of the crab fishery to the Introduction as advised (lines 121-124). We also provide additional text and interpretation of the spatial and temporal dynamics behind the enhanced conflict during the heatwave (lines 334-340, 354-367, 381-384).

Moreover, the Introduction concludes with the statement that the authors: “(1) evaluate social consequences of this human-wildlife conflict, (2) quantify the spatial and temporal dynamics behind them, and (3) explore the potential for dynamic management strategies to mitigate tradeoffs between whale conservation goals and the sustainability of the California Dungeness crab fishery before, during, and after this period of social-ecological squeeze”. From my reading point 3 is most clearly covered in the Discussion, but the others are not to the same extent. Statements made in the Discussion should also be more consistently supported with references, possibly to the grey literature, particularly for pages 9-10, paragraphs 3-2.

We updated paragraph 3 on p9 and paragraph 2 of p10 of the original ms to include multiple references to grey and peer-reviewed literature (revised ms lines 325-367). Furthermore, we threaded in Discussion of Figs. 1-3 throughout the first half of the Discussion, which allowed us to provide additional text and interpretation of the spatial and temporal dynamics behind the enhanced conflict during the heatwave. We believe that there is extensive discussion of the social consequences of the conflict (lines 319-324, 336-397), in terms of impacts on the crab fishery (revenue for the entire fishery and for small vessels only, which begins to address distributional impacts), the fact that ambiguity in societal values placed on conservation versus fisheries challenges the ability of decisionmakers to resolve conflicts, and where we encourage the development of unconventional and novel thinking on management solutions (e.g., incentive-based approaches).

Minor comments

Abstract: It would be good to specify the marine heatwave that is discussed in the manuscript here.

It would be useful to indicate which of the groups are the “losers” in the win-lose outcomes.

In the revised Abstract, we modified the sentences referring to the record heatwave and to winners and losers to read as follows: “We examined how the record 2014-16 Northeast Pacific marine heatwave influenced tradeoffs in managing conflict between conservation goals and human activities using a case study on large whale entanglements in the U.S. west coast’s most lucrative fishery (the Dungeness crab fishery). We showed that this extreme climate event diminished the power of multiple management strategies to resolve tradeoffs between entanglement risk and fishery revenue, transforming near win-win to clear win-lose outcomes (for whales and fishers, respectively).”

Page 4, paragraph 1: Add a reference for the definition of heatwaves.

Added.

Page 4, paragraph 2: Provide a reference and definition of marine heatwaves specifically – a commonly used definition can be found in:

Hobday, A. J., Alexander, L. V., Perkins, S. E., Smale, D. A., Straub, S. C., Oliver, E. C. J., Benthuisen, J. A., Burrows, M. T., Donat, M. G., Feng, M., Holbrook, N. J., Moore, P. J., Scannell, H. A., Sen Gupta, A., & Wernberg, T. (2016). A hierarchical approach to defining marine heatwaves. *Progress in Oceanography*, 141, 227–238.

<https://doi.org/10.1016/j.pocean.2015.12.014>

Or, alternatively:

Hobday, A. J., Oliver, E., Sen Gupta, A., Benthuisen, J., Burrows, M., Donat, M., Holbrook, N., Moore, P., Thomsen, M., Wernberg, T., & Smale, D. (2018). Categorizing and Naming Marine Heatwaves. *Oceanography*, 31(2). <https://doi.org/10.5670/oceanog.2018.205>

Oliver, E. C. J., Donat, M. G., Burrows, M. T., Moore, P. J., Smale, D. A., Alexander, L. V., Benthuisen, J. A., Feng, M., Sen Gupta, A., Hobday, A. J., Holbrook, N. J., Perkins-Kirkpatrick, S. E., Scannell, H. A., Straub, S. C., & Wernberg, T. (2018). Longer and more frequent marine heatwaves over the past century. *Nature Communications*, 9(1), 1324.

<https://doi.org/10.1038/s41467-018-03732-9>

Added the following sentence: “A marine heatwave is defined as a ‘prolonged discrete anomalously warm water event that can be described by its duration, intensity, rate of evolution, and spatial extent’ [12].”

Page 4, paragraph 2: It is noted that in “many regions around the world large whales have experienced changes in the timing and pathways of migration and occurrence on feeding grounds”. It would be good to indicate if these changes are driven by heatwaves specifically, or due to other changes (e.g. ocean warming, acidification, pollution, short-term upwelling, etc), and to provide supporting references.

We revised this sentence to read: “Some of these shifts are due to heatwaves specifically [2,18], though not all are [19,20], and can be especially problematic when altered spatial distributions or movements lead to new or increased conflict with human activities, including collisions with ships [19,21] and escalation of incidental catch in fisheries (i.e., bycatch [22]). ”

Pages 4-5: It is noted that “changing environmental conditions, higher whale abundance and distributional shifts, and an increasingly crowded ocean” contribute to challenging conservation of whales. A case study is then discussed that is intended to highlight these points, but to me it isn't clear how these three aspects are exemplified (e.g. there is no discussion of how whale shifts have contributed).

We appreciate the reviewer's lens on how we introduce and motivate the work, and admit that there is some art to motivating concern and interest in this problem while also allowing space to fully explore it using the Methods and Results described later in the ms. Nonetheless, the revised Introduction (lines 97-127) more fully articulates the conclusions from previous work in this study system to clarify the three aspects mentioned above by the reviewer, rather than delaying the dissemination of that information until the Results section (as in the original version of the ms).

Page 5, paragraph 2: In the win-lose outcome, is it the protected species or fisheries that are winning/losing?

It could be either the protected species or the fishery. We recognize the original wording was ambiguous and have revised the sentence to read: “Dynamic ocean management to reduce bycatch of migratory and highly mobile species of conservation concern offers particular promise [35,36], but it remains an open question whether these strategies can consistently produce win-win outcomes for protected species (by reducing bycatch) and fisheries (by maintaining or increasing yields), or will at times result in win-lose or even lose-lose outcomes [37].”

Referee: 2

Comments to the Author(s)

This is a rich, and well-written study touching on at least two important topics: first, how a specific marine heat wave in the western U.S. affected a specific—albeit important—human-wildlife conflict (Dungeness crab fishing and whale entanglement); and second, how climate extremes might affect human wildlife-conflict and the availability of win-wins (or near win-wins) generally. It should be published with some minor-to-medium revisions.

We appreciate the reviewer's critical and constructive feedback.

I think the first of these topics—the crab-whale conflict—is covered very well. The authors compellingly show and explain how the heat wave changed this specific conflict by altering timing and distributions of fishing effort and whales.

However, I think the authors should dig into the second topic a bit more in the discussion and overall framing of the article. Specifically, it would be good to give the reader a more general sense, conceptually, of how climate extremes should be expected to affect human-wildlife conflicts. For instance, I can think of two general types of effects that climate extremes could have, one of which I would always expect to exacerbate the conflict, as in this particular case, but the other where you might see variation on a case-by-case basis.

Type 1: the extreme stresses both the human and wildlife system. Adding stress to both objectives moves the efficiency frontier towards the origin, which should make win-wins harder to find. To what extent does this mechanism apply here?

Type 2: the extreme moves species and fishing effort around in space and time. This seems like it shouldn't always make the conflict worse. In the whale-crab case it did (by making them coincide more), but I don't see why they should always do that (but if the authors think there is a reason, that would be interesting and worth stating).

The authors don't necessarily need to use these classifications, but they should describe, at a conceptual level, how generalizable they expect climate extremes exacerbating human-wildlife conflict to be in other systems, and why that is.

This perspective is an excellent way to broaden the focus of our paper, and the revised ms embraces it to include conceptualization of the potential for climate extremes to create both opportunities to reduce human-wildlife conflict as well to exacerbate it. We have added language to both the Introduction (see especially the last paragraph) and Discussion (see especially the first paragraph), along with references, to better frame these alternative outcomes conceptually (lines 129-133, 310-322).

Minor comments:

- Figure 3 caption: please remove tacked changes.

Removed.

- Is it possible to include points representing the status quo in Fig. 4?

We recognize that it is a bit difficult to see them but the yellow circles in the lower right of each panel represent the status quo (100% fishery revenue and 0% risk reduction in each year). We added this text to the caption for further clarity.

- Figure S2: why are the changes so much smaller and more uniform after the heat wave (2018-2019)? Is that related to the timing conflict with the migration not subsiding when the heatwave ended, as mentioned on page 8?

We appreciate the reviewer calling our attention to this issue. First, we want to remind the reviewer that the 2018-2019 period only involves 1 crab fishing season, from Nov 2018 until Jul 2019. As a result there are no error bars included in the figure for that time period. We flagged this issue in the figure caption of the original ms. Second, we note that expected changes in blue and humpback whale risk vary substantially across the scenarios we examined (from ~15% increases in risk to 15% reductions in risk), especially given that this period only includes one fishing season worth of spatial variation in fishing and whale dynamics. Last, we note that the crab season start was delayed in 2018-2019 in the northern region of California (due to product-quality and domoic acid concerns^{1,2}), and for a shorter time for some ports in the central region of California (also due to domoic acid concerns³). Due to perceived whale entanglement risk, the fishery also closed early on April 15, 2019, statewide⁴. The combined effects of domoic acid and product quality delays to the season start and early closure due to entanglement risk would lead to a more uniform influence of the alternative management scenarios we analyzed on whale risk and fishery revenue. While in the Results section of the original ms we stated “Estimated entanglement risk to blue and humpback whales ... declined in the 2019 season (when the fishery

¹ CDFW News. (Dec 21 2018). Northern California commercial Dungeness crab season delay extended. <https://cdfgnews.wordpress.com/2018/12/21/northern-california-commercial-dungeness-crab-season-delay-extended/>

² CDFW News. (Jan 7 2019). Northern commercial Dungeness crab season further delayed in ocean waters north of Patrick's Point, Humboldt County due to public health hazard. <https://cdfgnews.wordpress.com/2019/01/07/northern-commercial-dungeness-crab-season-further-delayed-in-ocean-waters-north-of-patricks-point-humboldt-county-due-to-public-health-hazard/>

³ CDFW News. (Dec 3 2018) Commercial Dungeness crab season to open in Sonoma County. <https://cdfgnews.wordpress.com/2018/12/03/commercial-dungeness-crab-season-to-open-in-sonoma-county/>

⁴ CDFW News. (Apr 2 2019). Commercial Dungeness crab season to close statewide April 15. <https://cdfgnews.wordpress.com/2019/04/02/commercial-dungeness-crab-season-to-close-statewide-april-15/>

closed early statewide, due to a legal settlement [30]).”, and in the Discussion section of the original ms we stated “The State has also instituted new regulations that essentially use a dynamic ocean management approach to address entanglement risk associated with the Dungeness crab fishery”, in the revised ms we have added the following sentence to the Results and Discussion sections to further clarify the result displayed in Fig S2:

- *Results: “We note that for the 2018-19 period, simulated management scenarios produced little change from the status quo compared to the pre-heatwave and heatwave periods (Figs. 3-4, S2).”*
- *Discussion: “Indeed, these regulations resulted in the implementation of a statewide fishery closure in mid-April of 2019 (causing a smaller and more uniform influence of the alternative management scenarios we simulated in our analysis post-heatwave; Figs. 3-4, S2).”*